# Role of *Helicobacter pylori* in Upper Gastrointestinal Bleeding Among Ischemic Stroke Hospitalizations: A Nationwide Study of Outcomes

**Urvish K. Patel** [1],*, **Mihir Dave** [2], **Anusha Lekshminarayanan** [3], **Nidhi Patel** [4], **Abhishek Lunagariya** [1], **Vishal Jani** [1] **and Mandip S. Dhamoon** [5]

[1]  Department of Neurology, Creighton University School of Medicine, Omaha, NE 68122, USA
[2]  Department of Internal Medicine, University of Nevada Reno School of Medicine, Las Vegas, NV 89102, USA
[3]  Department of Rehabilitation Medicine, Metropolitan Hospital Center, New York, NY 10029, USA
[4]  Department of Internal Medicine, Drexel University College of Medicine, Philadelphia, PA 19129, USA
[5]  Department of Neurology, Icahn School of Medicine at Mount Sinai, New York, NY 10029, USA
*  Correspondence: dr.urvish.patel@gmail.com; Tel.: +1-(201)-936-6715

**Abstract:** Introduction: *Helicobacter pylori* (*H. pylori*) is a well-recognized risk factor for upper gastrointestinal bleeding (UGIB). The exposure to tissue plasminogen activator (tPA), anti-platelets, and anticoagulants increases the risk of UGIB in acute ischemic stroke (AIS) patients, the risk stratification of *H. pylori* infection is not known. In this retrospective cross-sectional study, we aimed to evaluate the relationship between *H. pylori* and GIB in patients hospitalized with AIS. Methods: In the nationwide data, hospitalization for AIS was identified by primary diagnosis using International Classification of Diseases, clinical modification (ICD-9-CM) codes. Subgroup of patients with GIB and *H. pylori* were identified in AIS cohort. A stepwise multivariable logistic regression model was fitted to evaluate the outcome of upper GIB and role of *H. Pylori* in UGIB. Results: Overall 4,224,924 AIS hospitalizations were identified, out of which 18,629 (0.44%) had UGIB and 3122 (0.07%) had *H. pylori*. The prevalence of *H. pylori*-induced UGIB among UGIB in AIS was 3.05%. The prevalence of UGIB was markedly elevated among the *H. pylori* infection group (18.23% vs. 0.43%; *p* < 0.0001) compared to the non-*H. pylori* group. In multivariable regression analysis, *H. pylori* was associated with markedly elevated odds of UGIB (aOR:27.75; 95%CI: 21.07–36.55; *p* < 0.0001). Conclusion: *H. pylori* infection had increased risk-adjusted occurrence of UGIB amongst the AIS hospitalized patients. *H. pylori* testing may improve risk stratification for UGIB and lower the health care cost burden in stroke hospitalization.

**Keywords:** acute ischemic stroke; helicobacter pylori; gastrointestinal hemorrhage; tissue plasminogen activator; nationwide inpatient sample; gastrointestinal bleeding

## 1. Introduction

Patients with acute ischemic stroke (AIS) have various complications, including an estimated 1–5% incidence of gastrointestinal bleeding (GIB) [1–3]. Multiple risk factors are associated with GIB post-stroke, including older age, presence of underlying pathology, past history of peptic ulcer, history of hemorrhage, stroke severity at admission, and increased exposure to tissue plasminogen activator (tPA), antiplatelets, and anticoagulants [2,3]. The most common cause of upper GIB (UGIB) is peptic ulcer followed by less common causes such as esophagitis, erosive gastritis, duodenitis, Mallory–Weiss syndrome, angiodysplasia, neoplasm and Dieulafoy's lesion [4–7]. The most common causes of lower GIB are diverticulosis, hemorrhoids, angiodysplasia, inflammatory disease, and cancer in descending frequency [8,9]. According to an international, prospective study called the Reduction of

Atherothrombosis for Continued Health (REACH) Registry, >68,000 patients across six continents with established atherothrombotic or three atherothrombotic risk factors, 70% of 25,668 outpatients from 1599 practices in US have been treated with aspirin, 13% were given aspirin and antiplatelets, 8% aspirin and oral anticoagulants, and half among those who were not on aspirin were on anticoagulants or antiplatelet drugs [4]. Aspirin and antiplatelets cause ulcers and erosion, which increases risk of GIB, while anticoagulants exacerbate bleeding from existing gastrointestinal lesions [10]. The impact of aspirin and anticoagulants on outcomes is not well-characterized [11–15]. Bleeding can lead to hemodynamic instability, and antithrombotic may need to be discontinued, leading to a prothrombotic state [16]. GIB has led to an increased hospital length of stay and poorer neurological prognosis, consequently having a significant impact on morbidity and mortality [3].

If potential risk factors for GIB in stroke can be identified, these could be used to reduce risk. One of the major risk factors for UGIB is peptic ulcer disease (PUD). *Helicobacter pylori* (*H. pylori*) and non-steroidal anti-inflammatory drug (NSAID) usage are two of the most common risk factors of PUD [17–21]. *H. pylori* is a gram-negative, spiral-shaped microaerophilic bacterium, colonizing the gastric mucosa that has been associated seroepidemiologically with atherosclerosis [22]. *H. pylori* may be a lifelong bacterial infection. Proton pump inhibitors and H2 antagonists reduce the risk of GIB and can be used in stroke patients who are at greater risk; hence, identifying risk factors for GIB could play a vital role in the management of stroke patients [22]. However, data regarding GIB in AIS are limited.

In this retrospective cross-sectional study, we aimed to ascertain the burden of GIB in patients hospitalized for AIS with serological evidence of *H. pylori* infection and to assess the risk of GIB with chronic use of aspirin and NSAIDs, and treatment with aspirin, antiplatelets, anticoagulants and tPA. We used the nationally representative Nationwide Inpatient Sample (NIS) from 2003 to 2014.

## 2. Materials and Methods

We had obtained the data from the Agency for Healthcare Research and Quality's Healthcare Cost and Utilization Project (HCUP) NIS files between January 2003 and December 2014. The NIS is a deidentified, the publicly available, largest inpatient care database in the USA and contains discharge-level data provided by 46 states that participate in the HCUP. This administrative dataset contains data of approximate a 20% stratified sample of all US hospitals which representing more than 95% of the national population. Discharge weight are provided to calculate the national estimate. For each hospitalization, an individual entry is available with one principle diagnosis and each hospitalization has up to 24 secondary diagnosis and 15 procedural diagnosis. Detailed information on NIS is available at http://www.hcup-us.ahrq.gov/db/nation/nis/nisdde.jsp.

### 2.1. Study Population

We used the 9th revision of the International Classification of Diseases, clinical modification codes (ICD-9-CM) to identify adult patients admitted with a primary diagnosis of AIS (ICD-9-CM codes 433.01, 433.11, 433.21, 433.31, 433.81, 433.91, 434.01, 434.11, 434.91). These codes have been previously validated and are 35% sensitive, 99% specific, 96% positive predictive value (PPVs), and 79% negative predictive value for the diagnosis of ischemic stroke [23]. Similarly, Patients with UGIB were identified using ICD-9-CM codes 530.xx-535.xx (PPVs: 66% for 532.xx, 61% for 531.xx, 534xx, and 1% for 533.xx) [24] and *H. pylori* using ICD-9-CM code 041.86 (PPVs: 100%) [25]. We used ICD-9-CM codes to identify independent predictors (covariates), including the comorbidities of hypertension, diabetes mellitus, hypercholesterolemia, atrial fibrillation, use of anticoagulant and antiplatelet medications, chronic use of NSAIDs and aspirin, smoking (current/past), and use of IV tPA. Supplementary Table S1 lists all ICD-9-CM codes that were used for this study.

Patients were divided into UGIB subgroups stratified by *H. pylori* status. Age < 18 years and admissions with missing data for age, sex, and race were excluded.

### 2.2. Patient and Hospital Characteristics

Patient characteristics of interest were sex, age, race, insurance status and concomitant diagnoses as defined above. Race was defined by white (referent), African American, Hispanic, Asian or Pacific Islander, and Native American. Insurance status was defined by Medicare (referent), Medicaid, Private Insurance, and Other/Self-pay/No charge. Severity of co-morbid conditions were defined by using Deyo's modification of the Charlson co-morbidity index (Supplementary Table S2). NIS data covered thirty-one facilities to be teaching hospitals if they have an American Medical Association-approved residency program, and are a member of the Council of Teaching Hospitals.

### 2.3. Outcomes

Our primary outcome of interest was UGIB with *H. pylori*, and secondary outcomes were death during hospitalization, All Patients Refined Diagnosis Related Groups (APRDRG) Risk of Mortality, APRDRG Severity estimate, discharge disposition, length of stay (LoS), and cost of hospitalization.

### 2.4. Statistical Analysis

We used SAS (version 9.4) for the all statistical analyses. From NIS data, weighted values of patient-level observations were obtained to produce a nationally representative estimate of hospitalized patients from the entire US population. We had considered $p$-values of $< 0.05$ as significant. We used chi-square test to evaluate the differences between categorical variables and paired student's $t$-test to analyse the differences between continuous variables including age, length of stay and cost of hospitalization. Hierarchical mixed-effects survey logistic regression models with weighted analysis were used for the categorical dependent variables, including UGIB and *H. Pylori* infection, in order to estimate odds ratio (OR) and 95% confidence interval for the association between UGIB and *H. Pylori*.

Three-level hierarchical models (demographics and patient-level factors nested within hospital-level factors) were created as random effects within the model for the primary outcome. In the multivariate analysis, we included demographics (age, gender, race), patient-level hospitalization variables (admission day, primary payer, admission type, Median Household Income Category), hospital-level variables (hospital region, teaching versus nonteaching hospital, hospital bed size), comorbidities (mentioned in Table 1), concurrent conditions including hypertension, diabetes mellitus, hypercholesterolemia, atrial fibrillation, obesity, hemorrhagic conversion, smoking status, drug abuse, alcohol abuse, medication use (anticoagulant and antiplatelet medication, chronic use of NSAIDs and aspirin), use of IV tPA during the same hospitalization or in a different institution within the 24 h prior to admission to the facility, and Charlson's co-morbidity index (CCI).

We investigated the variation in UGIB due to *H. pylori* by creating three separate stepwise hierarchical survey logistic regression models with weights to account for sampling strategy:

Model 1: Model with *H. pylori*.

Model 2: Model 1 + basic demographic-adjusted model including age, race, and sex with patient-level variables: admission day, primary payer and urgent/emergent admission, comorbidities, CCI, concurrent conditions like hypertension, diabetes mellitus, hypercholesterolemia, atrial fibrillation, obesity, hemorrhagic conversion, smoking status, drug abuse, alcohol abuse, medication use (anticoagulant and antiplatelet medication, chronic use of NSAIDs and aspirin), and use of IV tPA during the same hospitalization or in a different institution within the 24 hours prior to admission to the facility.

Model 3: Model 2 + hospital-associated characteristics including hospital region, teaching status, and bed size.

For each model, c-index was calculated to validate the accuracy of the regressions. We used two-sided tests with considering $p < 0.05$ statistically significant.

**Table 1.** Characteristics of *H. pylori* in upper GI bleeding among acute ischemic stroke population.

| | *H. pylori* | | | |
| --- | --- | --- | --- | --- |
| | **Yes** | **No** | **Total** | ***p* Value** |
| ***H. pylori* in Upper GI Bleeding (%)** | 569 (18.23) | 18,060 (0.43) | 18629 | <0.0001 |
| **Demographics of Patients** | | | | |
| **Mean Age ± Standard Error (Years)** | 71 ± 1.3 | 72 ± 0.2 | | 0.364 |
| **Age (Years) (%)** | | | | 0.0008 |
| 18–34 | <10 | 132 (0.73) | 137 (0.74) | |
| 35–49 | 28 (4.91) | 997 (5.52) | 1025 (5.50) | |
| 50–64 | 172 (30.26) | 4078 (22.58) | 4250 (22.81) | |
| 65–79 | 184 (32.43) | 6686 (37.02) | 6870 (36.88) | |
| ≥80 | 180 (31.60) | 6167 (34.15) | 6347 (34.07) | |
| **Gender (%)** | | | | <0.0001 |
| Female | 203 (35.70) | 8159 (45.18) | 8363 (44.89) | |
| Male | 366 (64.30) | 9900 (54.82) | 10266 (55.11) | |
| **Race (%)** | | | | <0.0001 |
| White | 266 (48.16) | 12,064 (68.50) | 12,330 (67.88) | |
| African American | 166 (29.95) | 3307 (18.78) | 3473 (19.12) | |
| Hispanic | 83 (14.93) | 1405 (7.98) | 1488 (8.19) | |
| Asian or Pacific Islander | 38 (6.96) | 743 (4.22) | 782 (4.30) | |
| Native American | 0 (0.00) | 92 (0.52) | 92 (0.51) | |
| **Characteristics of Patients** | | | | |
| **Median Household Income Category for patient's Zip code (%) \*** | | | | 0.0018 |
| 0–25th percentile | 201 (35.55) | 5310 (30.10) | 5511 (30.27) | |
| 26–50th percentile | 155 (27.52) | 4398 (24.93) | 4553 (25.01) | |
| 51–75th percentile | 112 (19.90) | 4127 (23.40) | 4239 (23.29) | |
| 76–100th percentile | 96 (17.03) | 3805 (21.57) | 3901 (21.43) | |
| **Primary Payer (%)** | | | | <0.0001 |
| Medicare | 347 (61.00) | 12,772 (70.87) | 13,119 (70.56) | |
| Medicaid | 52 (9.20) | 1397 (7.75) | 1449 (7.80) | |
| Private Insurance | 84 (14.74) | 2663 (14.78) | 2747 (14.78) | |
| Other/Self-pay/No charge | 86 (15.06) | 1191 (6.61) | 1277 (6.86) | |
| **Admission Type (%)** | | | | 0.1368 |
| Non- elective | 536 (94.19) | 17,231 (95.51) | 17,767 (95.47) | |
| Elective | 33 (5.81) | 810 (4.49) | 843 (4.53) | |
| **Admission Day (%)** | | | | 0.0541 |
| Weekday | 445 (78.18) | 13,475 (74.61) | 13,920 (74.72) | |
| Weekend | 124 (21.82) | 4585 (25.39) | 4709 (25.28) | |
| Characteristics of Hospitals | | | | |
| **Bedsize of Hospital (%) †** | | | | 0.6877 |
| Small | 51 (9.05) | 1801 (10.01) | 1852 (9.98) | |
| Medium | 139 (24.72) | 4541 (25.24) | 4680 (25.22) | |
| Large | 373 (66.24) | 11651 (64.75) | 12,024 (64.80) | |
| **Hospital Location & Teaching Status (%)** | | | | 0.3445 |
| Rural | 61 (10.80) | 1680 (9.34) | 1741 (9.38) | |
| Urban Non-teaching | 218 (38.68) | 7397 (41.11) | 7615 (41.04) | |
| Urban Teaching | 285 (50.52) | 8916 (49.55) | 9201 (49.58) | |
| **Hospital Region (%)** | | | | <0.0001 |
| Northeast | 96 (16.81) | 3662 (20.27) | 3757 (20.17) | |
| Midwest | 65 (11.36) | 2989 (16.55) | 3053 (16.39) | |
| South | 238 (41.88) | 7706 (42.67) | 7944 (42.65) | |
| West | 170 (29.95) | 3703 (20.50) | 3873 (20.79) | |
| **CM-Comorbidities of Patients (%)** | | | | |
| Diabetes Mellitus with/without complications | 218 (38.57) | 6017 (33.45) | 6235 (33.61) | 0.0114 |
| Drug abuse | 20 (3.49) | 411 (2.29) | 431 (2.32) | 0.0611 |
| Obesity | 24 (4.26) | 1049 (5.83) | 1073 (5.79) | 0.115 |
| Hypertension | 453 (80.36) | 13,121 (72.96) | 13,574 (73.18) | <0.0001 |
| Peripheral vascular disease | 63 (11.17) | 1820 (10.12) | 1883 (10.15) | 0.4133 |
| Renal failure | 107 (18.99) | 3266 (18.16) | 3374 (18.19) | 0.6172 |
| Coagulopathy | 15 (2.65) | 1293 (7.19) | 1308 (7.05) | <0.0001 |
| Solid tumor without metastasis | 25 (4.46) | 498 (2.77) | 523 (2.82) | 0.0166 |
| Other neurological disorders | <10 | 500 (2.78) | 505 (2.72) | 0.0044 |
| **Deyo's Charlson Comorbidity Index (CCI)** | | | | <0.0001 |
| 1 | 32 (5.60) | 2494 (13.81) | 2526 (13.56) | |
| 2 | 100 (17.62) | 3674 (20.34) | 3774 (20.26) | |
| 3 | 149 (26.24) | 3802 (21.05) | 3951 (21.21) | |
| 4 | 133 (23.33) | 3365 (18.63) | 3498 (18.77) | |
| ≥ 5 | 155 (27.21) | 4725 (26.16) | 4880 (26.19) | |

\* This represents a quartile classification of the estimated median household income of residents in the patient's ZIP Code. † Bedsize of hospital indicates number of hospital beds which varies depends on hospital location (Rural/Urban), teaching status (Teaching/Non-teaching) and Region (Northeast/Midwest/Southern/Western). Percentage in brackets are column % indicates direct comparison between *H. pylori* vs. Non-*H. pylori* amongst patients WITH Upper GI Bleeding.

## *2.5. Informed Consent*

The data has been taken from Nationwide Inpatient Sample, which is a deidentified database from "Health Care Utilization Project (HCUP)" sponsored by the Agency for Healthcare Research

and Quality, so informed consent or IRB approval was not needed for the study. The relevant ethical oversight and HCUP Data Use Agreement (HCUP-4Q28K90CU) were obtained for the data utilized in this study.

## 3. Results

### 3.1. Disease Hospitalizations

There were 4,224,924 hospitalizations due to AIS from 2003–2014 after excluding patients with age <18 years and admissions with missing data for age, gender, and race (Figure 1). Out of 4,224,924 hospitalizations, 18,629 (0.44%) had UGIB. Out of these 18,629 hospitalizations with AIS who had UGIB, 569 (3.05%) had concurrent *H. pylori* infection.

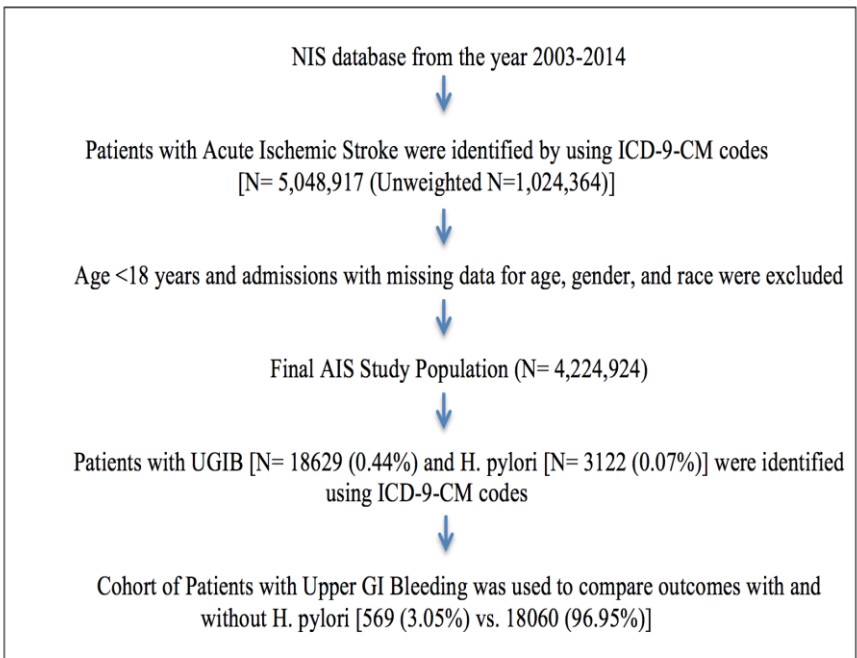

**Figure 1.** Flowchart detailing cohort selection and analysis modeling.

### 3.2. Prevalence Trends

We analyzed trends of UGIB and *H. pylori* in AIS hospitalizations. As shown in Figure 2, trends of UGIB in AIS hospitalizations and *H. pylori* infections in AIS hospitalizations were slightly declining from years 2003 to 2014. (UGIB: 0.61% in 2003 to 0.35% in 2014 and *H. pylori*: 0.12% in 2003 to 0.06% in 2014; P-Trend < 0.0001). However, the trend of *H. pylori* infection in UGIB drastically decreased from 3.75% in 2003 to 1.98% in 2014, being highest 4.23 in 2004; *p*-Trend < 0.0001.

### 3.3. Demographics, Patient and Hospital Characteristics, and Comorbidities

AIS hospitalizations with UGIB with concomitant *H. pylori* infection (compared to no infection) were more likely to be male (64.30% vs. 54.52%, *p* < 0.0001), African American (29.95% vs. 18.78%, *p* < 0.0001) and Hispanic (14.93% vs. 7.98%, *p* < 0.0001). Co-morbidities such as diabetes (38.57% vs. 33.45%, *p* = 0.0114) and hypertension (80.36% vs. 72.96%, *p* < 0.0001), peripheral vascular disease, and solid tumor without metastasis were more common in patients with *H. pylori* than those without *H. pylori* infection. Hospitalizations with *H. pylori* were also associated with a higher percentage of Deyo's Charlson Co-morbidity Index (CCI) 3, 4 and ≥5. AIS with UGIB hospitalizations in large, urban teaching hospitals and in the West region were more likely to have *H. pylori* infection (Table 1).

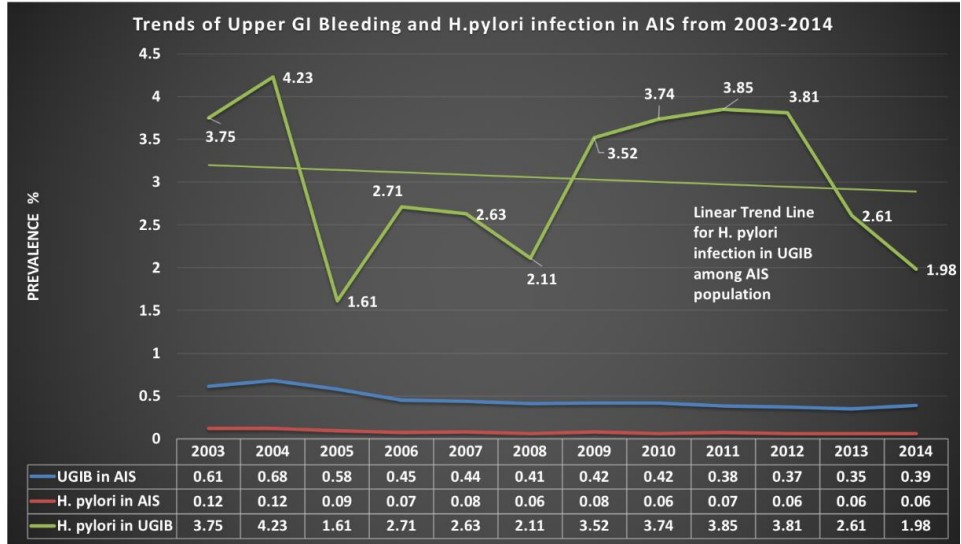

**Figure 2.** Prevalence trend of *H. pylori* infection.

*3.4. The Primary and Secondary Outcomes*

Table 2 shows outcomes of *H. pylori* infections in UGIB patients among AIS hospitalizations. UGIB was the primary outcome. Prevalence of *H. pylori* among UGIB was 18.23% vs. 0.43%, $p < 0.0001$) without *H. pylori*. Secondary outcomes were all-cause in-hospital mortality, discharge disposition, loss of function and risk of death. Though prevalence of all-cause in hospital mortality, major/extreme likelihood of death on discharge, and discharge outcomes were not higher among UGIB with *H. pylori*, the prevalence of major loss of function was higher (65.91% vs. 52.62%, $p < 0.0001$).

**Table 2.** Univariate analysis of outcomes of *H. pylori* infection.

| | *H. pylori* | | | |
| --- | --- | --- | --- | --- |
| | **Yes** | **No** | **Total** | *p* **Value** |
| **Primary Outcome: Upper GI Bleeding among AIS patients** | | | | |
| **Patients WITHOUT Upper GI Bleeding** | 2553 (81.77) | 4,203,743 (99.57) | 4,206,295 | |
| **Patients WITH Upper GI Bleeding** | 569 (18.23) | 18,060 (0.43) | 18,629 | <0.0001 |
| **Secondary Outcomes in Upper GI Bleeding Population** | | | | |
| All Cause in Hospital Mortality (%) | <10 | 2544 (14.12) | 2553 (13.74) | <0.0001 |
| **Discharge Disposition (%)** | | | | <0.0001 |
| Routine/Home | 184 (32.93) | 2693 (17.48) | 2877 (18.02) | |
| Transfer to Short-term Hospital | 32 (5.73) | 681 (4.42) | 713 (4.46) | |
| Transfer to SNF/ICF/Another Type of Facility | 264 (47.34) | 10,194 (66.15) | 10,458 (65.49) | |
| Home Health Care | 78 (14.01) | 1842 (11.95) | 1920 (12.03) | |
| Total Discharge other than Home (%) | 375 (67.07) | 12,716 (82.52) | 13,091 (81.98) | <0.0001 |
| **APRDRG Severity/Loss of Function (%)** | | | | <0.0001 |
| Minor loss of function | 0 (0) | 0 (0) | 0 (0) | |
| Moderate loss of function | 59 (10.59) | 1191 (7.73) | 1250 (7.83) | |
| Major loss of function | 366 (65.91) | 8108 (52.62) | 8474 (53.08) | |
| Severe loss of function | 130 (23.50) | 6109 (39.65) | 6240 (39.09) | |
| Total Major/Severe Loss of Function (%) | 496 (89.41) | 14,218 (92.27) | 14,713 (92.17) | 0.0138 |
| **APRDRG Likelihood of Death (%)** | | | | <0.0001 |
| Minor likelihood of death | 123 (22.12) | 1133 (7.35) | 1256 (7.87) | |
| Moderate likelihood of death | 224 (40.37) | 5234 (33.97) | 5458 (34.19) | |
| Major likelihood of death | 133 (23.97) | 5392 (34.99) | 5525 (34.61) | |
| Extreme likelihood of death | 75 (13.53) | 3650 (23.69) | 3725 (23.33) | |
| Total Major/Extreme likelihood of death (%) | 208 (37.50) | 9042 (58.68) | 9250 (57.94) | <0.0001 |
| **Length of Stay Mean ± SE (Days)** | 10.23 ± 0.69 | 12.77 ± 0.26 | | 0.0008 |
| **Cost of Hospitalization Mean ± SE ($)** | 68,945 ± 6834 | 94,812 ± 2421 | | 0.0005 |

APRDRG: All Patients Refined Diagnosis Related Groups; SNF: Skilled Nursing Facility; ICF: Intermediate Care Facility; SE: Standard Error; Percentage in brackets are column % indicates direct comparison between *H. pylori* vs. Non-*H. pylori* amongst patients WITH Upper GI Bleeding.

### 3.5. Length of Stay and Cost of Hospitalization

Mean length of stay (10.2 days vs. 12.8 days, *p* = 0.0008) and total cost of hospitalization were lower ($68,945 vs. $94,812, *p* = 0.0005) among those with *H. pylori* infection (Table 2).

### 3.6. Regression Model Derivation

The overall unadjusted odds for UGIB in *H. pylori* infections among AIS (model 1) were 52.33 (95%CI: 42.45–64.50, *p* < 0.0001), while after adjusting for basic demographic with patient-level variables, comorbidities, CCI, concurrent conditions in model 2, the adjusted odds for UGIB in *H. pylori* infection were 27.64 (95%CI: 20.99–27.40, *p* < 0.0001) compared to without *H. pylori* infection. Model 3 suggested, *H. pylori* was linked with UGIB with odds of 27.75 (95%CI: 21.07–36.55, *p* < 0.0001). (Table 3).

Table 3 also lists multivariate analysis of predictors of UGIB in AIS hospitalizations with *H. pylori* infections. Age group year 50–64 (compared to 18–34), male (compared to female) and African American (compare to White) race were significant predictors of UGIB with an adjusted OR of 1.50 (95%CI: 1.00–2.25, *p* = 0.0496), 1.39 (95%CI: 1.29–1.49, *p* < 0.0001), and 1.12 (95%CI: 1.01–1.23, *p* = 0.0248) respectively. Co-morbidities like diabetes (adjusted OR: 1.65, 95%CI: 1.02–2.65, *p* = 0.0409), obesity (aOR: 1.48, 95%CI: 1.04–2.11, *p* = 0.029), hypertension (aOR: 1.46, 95%CI: 1.24–1.71, *p* < 0.0001), coagulopathy (aOR: 1.85, 95%CI: 1.61–2.11, *p* < 0.0001), solid tumor without metastasis (aOR: 1.28, 95%CI: 1.04–1.56, *p* = 0.01888) and other neurological disorders (aOR: 3.31, 95%CI: 2.65–4.14, *p* < 0.0001) were significant predictors of UGIB with *H. pylori* infection amongst AIS hospitalizations. Also, more burden of comorbidities, CCI ≥ 5 (aOR: 4.85, 95%CI: 4.24–5.55, *p* < 0.0001) was associated with UGIB. Hospitals with more beds, urban hospital (teaching and nonteaching), and Western hospitals were associated with higher adjusted odds for UGIB.

Concurrent conditions/secondary diagnosis like atrial fibrillation (aOR: 1.12, 95%CI: 1.03–1.22, *p* = 0.0107), hemorrhagic conversion (aOR: 1.34, 95%CI: 1.1–1.64, *p* = 0.0045), alcohol abuse/dependence (aOR: 1.47, 95%CI: 1.26–1.72, *p* < 0.0001), and chronic use of NSAIDs (aOR: 3.19, 95%CI: 1.64–6.16, *p* = 0.0006) were also significant predictors of UGIB. Concurrent conditions like chronic (current) use of aspirin (aOR: 0.63, 95%CI: 0.54-0.74, *p* < 0.0001), current use of anti-platelet medicines (aOR: 0.66, 95%CI: 0.49–0.90, *p* = 0.0077), use of anti-coagulant medicines (aOR: 0.66, 95%CI: 0.55–0.78, *p* < 0.0001) had lower odds of UGIB.

We did not find that recombinant tissue plasminogen activator (t-PA) was associated with UGIB in patients with *H. pylori* (*p* = 0.2974).

**Table 3.** Multivariate logistic regression analysis for the outcome of upper GI bleeding.

| | Model 1 * | | | | Model 2 † | | | | Model 3 @ | | | |
|---|---|---|---|---|---|---|---|---|---|---|---|---|
| | OR | CI | | *p* value | OR | CI | | *p* Value | OR | CI | | *p* Value |
| | | LL | UL | | | LL | UL | | | LL | UL | |
| **No *H. pylori*** | | | | | | | Reference | | | | | |
| ***H. pylori*** | 52.33 | 42.45 | 64.5 | <0.0001 | 27.64 | 20.99 | 36.4 | <0.0001 | 27.75 | 21.07 | 36.55 | <0.0001 |
| **Age (Years)** | | | | | | | | | | | | |
| 18–34 | | | | | | | Reference | | | | | |
| 35–49 | | | | | 1.28 | 0.84 | 1.96 | 0.2475 | 1.31 | 0.86 | 1.99 | 0.2152 |
| 50–64 | | | | | 1.47 | 0.98 | 2.21 | 0.0629 | 1.5 | 1 | 2.25 | 0.0496 |
| 65–79 | | | | | 1.44 | 0.95 | 2.19 | 0,0833 | 1.48 | 0.98 | 2.24 | 0.0641 |
| ≥ 80 | | | | | 1.32 | 0.87 | 2 | 0.1935 | 1.37 | 0.9 | 2.07 | 0.1436 |
| **Gender** | | | | | | | | | | | | |
| Female | | | | | | | Reference | | | | | |
| Male | | | | | 1.39 | 1.29 | 1.49 | <0.0001 | 1.39 | 1.29 | 1.49 | <0.0001 |
| **Race** | | | | | | | | | | | | |
| White | | | | | | | Reference | | | | | |
| African American | | | | | 1.15 | 1.04 | 1.26 | 0.0049 | 1.12 | 1.01 | 1.23 | 0.0248 |
| Hispanic | | | | | 1.08 | 0.95 | 1.23 | 0.2243 | 1.03 | 0.91 | 1.17 | 0.65 |
| Asian or Pacific Islander | | | | | 1.6 | 1.35 | 1.9 | <0.0001 | 1.5 | 1.26 | 1.78 | <0.0001 |
| Native American | | | | | 1.11 | 0.67 | 1.83 | 0.6809 | 1.11 | 0.68 | 1.83 | 0.6764 |
| **Median Household Income Category for Patient's Zip Code** | | | | | | | | | | | | |
| 0–25th percentile | | | | | | | Reference | | | | | |
| 26–50th percentile | | | | | 1.01 | 0.93 | 1.11 | 0.7606 | 1.01 | 0.92 | 1.11 | 0.8095 |
| 51–75th percentile | | | | | 1.05 | 0.95 | 1.15 | 0.3506 | 1.03 | 0.93 | 1.13 | 0.6175 |
| 76–100th percentile | | | | | 1.07 | 0.97 | 1.18 | 0.1669 | 1.05 | 0.95 | 1.16 | 0.3728 |
| **Primary Payer** | | | | | | | | | | | | |
| Medicare | | | | | | | Reference | | | | | |
| Medicaid | | | | | 1.05 | 0.9 | 1.23 | 0.5592 | 1.03 | 0.88 | 1.21 | 0.678 |
| Private Insurance | | | | | 0.84 | 0.74 | 0.94 | 0.0033 | 0.83 | 0.73 | 0.93 | 0.0019 |
| Other/Self-pay/No charge | | | | | 1.02 | 0.87 | 1.2 | 0.8479 | 1 | 0.85 | 1.18 | 0.9779 |

**Table 3.** *Cont.*

| | Model 1 * | | | Model 2 † | | | | Model 3 @ | | | |
|---|---|---|---|---|---|---|---|---|---|---|---|
| | OR | CI | *p* value | OR | CI | | *p* Value | OR | CI | | *p* Value |
| | | LL | UL | | LL | UL | | | LL | UL | |
| **Admission Type** | | | | | | | | | | | |
| Non-elective | | | | | Reference | | | | | | |
| Elective | | | | 1 | 0.85 | 1.17 | 0.9516 | 1.02 | 0.87 | 1.2 | 0.7788 |
| **Admission Day** | | | | | | | | | | | |
| Weekday | | | | | Reference | | | | | | |
| Weekend | | | | 0.98 | 0.91 | 1.06 | 0.5769 | 0.98 | 0.9 | 1.05 | 0.542 |
| **CM- Comorbidities of Patients** | | | | | | | | | | | |
| Diabetes Mellitus with/without complications | | | | 1.69 | 1.05 | 2.73 | 0.0305 | 1.65 | 1.02 | 2.65 | 0.0409 |
| Drug abuse | | | | 1.53 | 0.48 | 4.89 | 0.4761 | 1.49 | 0.47 | 4.77 | 0.5032 |
| Obesity | | | | 1.5 | 1.05 | 2.14 | 0.0254 | 1.48 | 1.04 | 2.11 | 0.029 |
| Hypertension | | | | 1.48 | 1.26 | 1.73 | <0.0001 | 1.46 | 1.24 | 1.71 | <0.0001 |
| Peripheral vascular disease | | | | 0.94 | 0.84 | 1.05 | 0.2863 | 0.94 | 0.84 | 1.05 | 0.2525 |
| Renal failure | | | | 0.81 | 0.72 | 0.9 | 0.0001 | 0.81 | 0.72 | 0.9 | 0.0001 |
| Coagulopathy | | | | 1.87 | 1.64 | 2.14 | <0.0001 | 1.85 | 1.61 | 2.11 | <0.0001 |
| Solid tumor without metastasis | | | | 1.28 | 1.04 | 1.57 | 0.0188 | 1.28 | 1.04 | 1.57 | 0.0188 |
| Other neurological disorders | | | | 3.4 | 2.72 | 4.25 | <0.0001 | 3.31 | 2.65 | 4.14 | <0.0001 |
| **Concurrent Conditions/Secondary Diagnosis** | | | | | | | | | | | |
| Hyperlipidemia | | | | 0.83 | 0.73 | 0.94 | 0.0028 | 0.83 | 0.73 | 0.94 | 0.0031 |
| Atrial fibrillation | | | | 1.13 | 1.03 | 1.23 | 0.0067 | 1.12 | 1.03 | 1.22 | 0.0107 |
| Hemorrhagic conversion | | | | 1.38 | 1.13 | 1.69 | 0.0018 | 1.34 | 1.1 | 1.64 | 0.0045 |
| Alcohol abuse/dependence | | | | 1.47 | 1.25 | 1.72 | <0.0001 | 1.47 | 1.26 | 1.72 | <0.0001 |
| Tobacco current/past use | | | | 0.63 | 0.57 | 0.69 | <0.0001 | 0.62 | 0.57 | 0.68 | <0.0001 |
| Chronic use of NSAIDs | | | | 3.18 | 1.63 | 6.19 | 0.0007 | 3.19 | 1.64 | 6.21 | 0.0006 |
| Chronic use of Aspirin | | | | 0.64 | 0.55 | 0.75 | <0.0001 | 0.63 | 0.54 | 0.74 | <0.0001 |
| Use of Anti-platelet medicine | | | | 0.67 | 0.5 | 0.91 | 0.0093 | 0.66 | 0.49 | 0.9 | 0.0077 |
| Use of Anti-coagulant medicine | | | | 0.66 | 0.55 | 0.79 | <0.0001 | 0.66 | 0.55 | 0.78 | <0.0001 |
| Use of Recombinant tissue plasminogen activator (t-PA) | | | | 0.95 | 0.81 | 1.11 | 0.4881 | 0.92 | 0.79 | 1.08 | 0.2974 |

**Table 3.** *Cont.*

| | Model 1 * | | | Model 2 † | | | | Model 3 @ | | | |
|---|---|---|---|---|---|---|---|---|---|---|---|
| | OR | CI | *p* value | OR | CI | | *p* Value | OR | CI | | *p* Value |
| | | LL | UL | | LL | UL | | | LL | UL | |
| **Deyo's Charlson Comorbidity Index (CCI)** | | | | | | | | | | | |
| 1 | | | | | Reference | | | | | | |
| 2 | | | | 2.29 | 2.01 | 2.6 | <0.0001 | 2.29 | 2.02 | 2.6 | <0.0001 |
| 3 | | | | 2.23 | 1.99 | 2.51 | <0.0001 | 2.22 | 1.98 | 2.5 | <0.0001 |
| 4 | | | | 3.32 | 2.91 | 3.78 | <0.0001 | 3.3 | 2.88 | 3.75 | <0.0001 |
| ≥ 5 | | | | 4.91 | 4.29 | 5.62 | <0.0001 | 4.85 | 4.24 | 5.55 | <0.0001 |
| **Bedsize of Hospital (# of beds in hospital)** | | | | | | | | | | | |
| Small | | | | | Reference | | | | | | |
| Medium | | | | | | | | 1.1 | 0.97 | 1.24 | 0.1411 |
| Large | | | | | | | | 1.15 | 1.03 | 1.29 | 0.0135 |
| **Hospital Location & Teaching Status** | | | | | | | | | | | |
| Rural | | | | | Reference | | | | | | |
| Urban Non-teaching | | | | | | | | 1.17 | 1.03 | 1.33 | 0.0175 |
| Urban Teaching | | | | | | | | 1.31 | 1.15 | 1.48 | <0.0001 |
| **Hospital Region** | | | | | | | | | | | |
| Northeast | | | | | Reference | | | | | | |
| Midwest | | | | | | | | 1.08 | 0.96 | 1.21 | 0.2198 |
| South | | | | | | | | 1.11 | 1.01 | 1.22 | 0.0322 |
| West | | | | | | | | 1.2 | 1.08 | 1.34 | 0.001 |
| **c-index** | **0.515** | | | **0.705** | | | | **0.706** | | | |

OR: Odds Ratio; CI: Confidence Interval; UL: Upper Limit; LL: Lower Limit. * Model 1: Upper GI Bleeding = *H. pylori.* † Model 2: model 1 + basic demographic with patient-level variables, comorbidities, CCI, concurrent conditions. @ Model 3: model 2 + hospital-level variables such as hospital region, teaching status, and bed size.

*3.7. Accuracy of the Model:*

The c-index was 0.515, 0.705, 0.706 for unadjusted model 1, adjusted model 2 and adjusted model 3, respectively. Both adjusted model 2 and 3 have c-index > 0.7, which indicates a good model fit.

## 4. Discussion

Upper gastrointestinal bleeding (UGIB) is a common complication following an acute ischemic stroke, with an incidence ranging from 1–5% [1–3]. It is known that *H. pylori* serves as a positive risk factor for peptic ulcer disease (PUD), which could ultimately lead to UGIB [17–21]. Our results suggest that *H. pylori* was significantly associated with UGIB (adjusted odds ratio: 27.75), possibly due to the relationship between *H. pylori* and PUD. The study also significantly found that hospitalizations with UGIB with concomitant *H. pylori* infection were more likely to be male, African American and Hispanic.

*H. pylori* increases the risk of PUD through a number of virulence factors which influence its colonization and severity. In particular, cytotoxin-associated gene A (cagA), encodes a type IV secretion apparatus, which is used to inject CagA, which is thought to interact with multiple host proteins, into the host cell cytoplasm to cause inflammation and increased risk of ulcers. Similarly, all *H. Pylori* strains contain vacuolating toxin A (VacA), which produce vacA genes. The strains of *H. pylori* harboring the s1m1 allele and i1 allele are thought to also play a substantial role in the development of ulcers, due to their high toxicity [26]. Additionally, *H. pylori* gamma-glutamyl transpeptidase (GGT) also functions to increase peptic ulcer development through increasing the secretion of IL-8 and hydrogen peroxide in epithelial cells [26]. These mechanisms contribute to the ability of *H. pylori* to contribute to the formation of PUD, however, the percentage of ulcers not associated with *H. pylori* is rising due to increased use of NSAIDs.

PUD is significantly associated with the use of NSAIDs. In our study, we found that chronic use of NSAIDs was a significant predictor of UGIB. In a meta-analysis study done on the relationship between *H. pylori* infection and the use of NSAIDs in the pathogenesis of PUD, it was found that PUD was significantly more common in NSAID takers, irrespective of *H. pylori* infection and the use of NSAIDs increased the risk of ulcer bleeding in the presence of *H. pylori* infection [27]. However, previous studies have also demonstrated that *H. pylori* eradication is partially protective against ulcer development in NSAID users [28,29]. This demonstrates that although NSAID usage and *H. pylori* infection could be independently associated with the development of PUD, there is also some degree of overlap between them in the pathogenesis of PUD. Our study also found that chronic use of aspirin had lower odds of UGIB. However, this finding contradicts previous studies which found that aspirin doubles the risk of bleeding, even at doses as low as 75 mg daily [30]. In our study major risk factors for the upper GI events were alcohol consumption, drug, hypertension, which in presence of emotional stress could also contribute to ulcer formation by increasing secretion of gastric acid and damaging the mucosal barrier [31].

In one study, it was found that *H. pylori* is treated relatively poorly in inpatients compared to outpatients and that inpatients received therapy only 60% of the time when compared to outpatients, who received therapy 92% of the time [32]. This finding could also be attributed to relatively poor treatment of inpatients with *H. pylori* infection. The mean length of stay and cost of hospitalization could have been lower due to patients being discharged early, which could be due to the lack of appropriate care in hospitals.

Our study also showed utilization of IV-tPA had no significant role in UGIB among patients with or without *H. pylori* infection.

The main strength of this study is the number of hospitalizations from NIS database. The nationwide data is having inpatient records of approximately 9–12 million patients each year, which makes our study results more reliable. With the ICD-9-CM codes, we utilized have high PPV of stroke, *H. pylori*, and UGIB [23–25]. Thirumurthi et al. utilized serology and urea breath test to identify *H. pylori* infection and to compare PPV of ICD-9-CM code for in-patient and outpatient cohort [25]. The sensitivity and specificity of urea breath test are approximately 88–95% and 95–100%,

respectively [33] and sensitivity and specificity of serologic assays found an overall of 85% and 79%, respectively to differentiate an active vs. past infection [34]. However, this study has several limitations. This is a retrospective cross-sectional study based on patients hospitalized for AIS. Due to the nature of this study, selection biases could exist. Sensitivity, specificity and PPV for other confounders and comorbidities could not be measured accurately in the NIS database. Additionally, data from clinical registries or administrative databases were obtained retrospectively or prospectively by chart abstractions based on the discharge diagnosis codes, billing codes etc. and hence are susceptible to coding errors.

## 5. Conclusions

These results indicate that the *H. pylori* infection is highly associated with UGIB amongst the patients with AIS. Believed risk factors like utilization of aspirin, anticoagulants, antiplatelets, and IV tPA have no role in increasing the risk of UGIB. Early identification and treatment of *H. pylori* infection amongst AIS patients improve the outcomes and risk stratification for GIB. These results emphasize the need for a hospital-based randomized study to evaluate the benefit of *H. pylori* infection identification and eradication in stroke patients and to track them to see the risk of GIB in future.

**Supplementary Materials:** The following are available online at http://www.mdpi.com/2624-5647/1/3/29/s1. Table S1: ICD-9-CM codes used in this analysis; Table S2: Deyo's modification of Charlson's co-morbidity index (CCI).

**Author Contributions:** Conceptualization, U.K.P.; methodology, U.K.P., M.S.D.; software, U.K.P.; validation, M.S.D., V.J., A.L. (Abhishek Lunagariya); formal analysis, U.K.P.; investigation, U.K.P.; resources, M.S.D.; data curation, M.D.; writing—original draft preparation, U.K.P., A.L. (Anusha Lekshminarayanan), M.D., N.P.; writing—review and editing, M.S.D., A.L. (Abhishek Lunagariya), V.J.; visualization, U.K.P.; supervision, M.S.D.; project administration, U.K.P.; funding acquisition, none.

**Funding:** This research received no external funding.

**Conflicts of Interest:** The authors declare no conflict of interest.

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
