# Peer review of "Role of Helicobacter pylori in Upper Gastrointestinal Bleeding Among Ischemic Stroke Hospitalizations: A Nationwide Study of Outcomes"

_gastrointestdisord, doi:10.3390/gidisord1030029_

Round 1

Reviewer 1 Report

In this retrospective cross-sectional study, we aimed to ascertain the burden of GIB in patients 73 hospitalized for AIS with serological evidence of H. pylori infection and to assess the risk of GIB with 74 chronic use of aspirin and NSAIDs, and treatment with aspirin, antiplatelets, anticoagulants and tPA.

The authors, correctly, reported that the study design could be a limitation. I think that another limitation should be described: the fact that serology could not represent an active Helicobacter pylori infection but a previous contact with the bacterium.

Author Response

Dear Reviewer-

I appreciate your effort to make literature more accurate and I have noted your comment and will modify the draft accordingly.

Thank you,

Sincerely,

Urvish

Reviewer 2 Report

Comments to the Authors

The manuscript from Urvish K. Patel et al. entitled “Role of Helicobacter Pylori in Upper Gastrointestinal Bleeding Among Ischemic Stroke Hospitalizations: A Nationwide Study of Outcomes” seems to be an interesting manuscript. There are points which need to be addressed.

Some numbers have commas, others do not. For example, 4,224,924 (Line 31) and 18629 (Line 32). Authors recommend they all have. The words “Helicobacter pylori” are spelled in several different ways. Also, authors should check if a bacterial name must be in italics. Reviewer recommend the words are abbreviated as “HP.” Line 170, “CCI (Deyo’s Charlson Co-morbidity) Index” seems strange, should be “Deyo’s Charlson Co-morbidity Index (CCI).” In Discussion, there are many words “peptic ulcer disease” and “PUD.” “Upper gastrointestinal bleeding” is also already described and should be abbreviated. Line 259, readers cannot understand the sentence “our study also showed utilization if (correct word should be of) IV-tPA had no significant role in UGIB among patients with or without H. pylori infection.”

Author Response

Dear Reviewer-

I appreciate your time and effort to make literature accurate and reader-friendly.

I have corrected the draft according to the suggestions.

Thank you

Sincerely,

Urvish